# Upsetting the Balance: How Modifiable Risk Factors Contribute to the Progression of Alzheimer’s Disease

**DOI:** 10.3390/biom14030274

**Published:** 2024-02-24

**Authors:** Caitlin M. Carroll, Ruth M. Benca

**Affiliations:** 1Psychiatry and Behavioral Medicine, Wake Forest University School of Medicine, Winston-Salem, NC 27157, USA; cmcarrol@wakehealth.edu; 2Alzheimer’s Disease Research Center, Wake Forest University School of Medicine, Winston-Salem, NC 27157, USA

**Keywords:** Alzheimer’s disease, sleep, metabolism, aging, amyloid-beta (Aβ), lifestyle interventions

## Abstract

Alzheimer’s disease (AD) is a neurodegenerative disorder affecting nearly one in nine older adults in the US. This number is expected to grow exponentially, thereby increasing stress on caregivers and health systems. While some risk factors for developing AD are genetic, an estimated 1/3 of AD cases are attributed to lifestyle. Many of these risk factors emerge decades before clinical symptoms of AD are detected, and targeting them may offer more efficacious strategies for slowing or preventing disease progression. This review will focus on two common risk factors for AD, metabolic dysfunction and sleep impairments, and discuss potential mechanisms underlying their relationship to AD pathophysiology. Both sleep and metabolism can alter AD-related protein production and clearance, contributing to an imbalance that drives AD progression. Additionally, these risk factors have bidirectional relationships with AD, where the presence of AD-related pathology can further disrupt sleep and worsen metabolic functioning. Sleep and metabolism also appear to have a bidirectional relationship with each other, indirectly exacerbating AD pathophysiology. Understanding the mechanisms involved in these relationships is critical for identifying new strategies to slow the AD cascade.

## 1. Alzheimer’s Disease Background

Alzheimer’s disease (AD) is a progressive disease, with biomarker changes detected decades before the onset of clinical symptoms. Currently, there are more than 55 million individuals worldwide living with dementia, costing more than 1.3 trillion USD annually [1]. As the population continues to age, these numbers are expected to grow exponentially, increasing the pressure to identify novel treatments [1]. According to the proposed A/T/N framework [2,3], AD is defined by the presence of pathology within the following biomarker categories: intracellularly aggregated amyloid-beta (Aβ; “A”), extracellularly aggregated pathological tau (“T”), and neurodegeneration or neuronal injury (“N”). Changes in these biomarker profiles, along with declines in cognitive performance, indicate progression along the AD continuum. This pathological cascade, in which an individual may move from preclinical Alzheimer’s disease to Alzheimer’s disease with mild cognitive impairment (MCI) or dementia, likely occurs over decades [4]. Amyloid deposition occurs 15–20 years before clinical symptom onset [5,6], while increased CSF tau and neurodegeneration likely start 10–15 years before symptoms manifest [4,6]. Other biomarkers, such as altered brain glucose metabolism [6,7,8,9] and markers of inflammation [10,11], as well as modifiable risk factors for AD [12,13,14,15], emerge during this early preclinical phase and may even contribute to disease pathogenesis [8,16,17,18,19]. Therefore, the most clinically efficacious treatments should target this long pre-clinical period to slow biomarker accumulation and stop disease progression. 

The traditional amyloid cascade hypothesis posits that AD pathophysiology is spurred by an imbalance in Aβ production and clearance mechanisms, and recent studies suggest similar mechanisms may contribute to early tau hyperphosphorylation and spreading [20,21,22]. Aβ is produced under normal physiological circumstances as a by-product of neuronal activity through the sequential cleavage of amyloid precursor protein (APP) by β- and γ-secretase [23]. APP can also be cleaved by α- and γ-secretase in a non-amyloidogenic pathway, resulting in the release of non-plaque-forming extracellular peptides [24]. Genetic mutations in this APP processing pathway result in increased amyloidogenic pathway activity and are a common cause of early-onset Alzheimer’s disease [25,26,27]. In AD, Aβ aggregates in a concentration-dependent manner into insoluble amyloid plaques. Because of this, Aβ plaques have a regional distribution whereby highly active regions, such as the default mode network, show early and dense plaque accumulation [28,29]. Similarly, neuronal activity also regulates tau release [30,31] and propagation [32], suggesting targeting neuronal activity levels may simultaneously affect both AD pathologies. Aβ is regularly cleared through a variety of mechanisms, including intracellular (e.g., ubiquitin-proteasome and autophagy lysosome systems) and extracellular (e.g., glial phagocytosis) degradation processes [33,34]. Aβ can also be cleared into the periphery across the blood–brain barrier (BBB) [35] and through the glymphatic system, which facilitates Aβ clearance from interstitial fluid (ISF) via astrocytic aquaporin-4 (AQP-4) channels [36]. Studies have shown that impaired clearance mechanisms, such as diminished perivascular drainage across the BBB in cerebral small-vessel disease, lead to increased Aβ accumulation [37]. Recent studies have also demonstrated that the glymphatic system may play a role in tau clearance. Inhibition of AQP-4 function or deletion of the channel increased tau levels and facilitated neurodegeneration in mice [38,39,40]. These studies demonstrate that both overproduction and reduced clearance can cause an imbalance in the regulation of Aβ and tau, thereby increasing AD pathology [41,42,43]. Therefore, treatments targeting mechanisms of production and/or clearance should reduce the overall pathological load and slow AD progression.

There are currently few drugs approved for the treatment of AD pathology and symptoms. Cholinesterase inhibitors (e.g., Donepezil, Rivastigmine) and glutamate regulators (e.g., Memantine) are both approved to treat cognitive symptoms. While these drugs have a modest beneficial impact on functional outcomes, they have no effect on disease progression [44,45]. Recently, the FDA has approved several anti-amyloid therapies for the treatment of AD. These monoclonal antibody therapies are successful at reducing Aβ plaque burden in the brain [46]. However, there is still debate as to whether the clinical efficacy outweighs the potential side effect risks for these drugs, particularly with the first approved drug in this class, Aducanumab. However, results from the Lecanemab and Donanemab clinical trials showed significant slowing of cognitive decline, equating to around a 4–6 month delay in cognitive deterioration, particularly among those with a low disease burden, suggesting more functional efficacy in slowing disease progression [46,47]. However, these cognitive benefits are still coupled with a relatively large risk of monoclonal antibody-related side effects, namely amyloid-related imaging abnormalities (ARIA). ARIA can cause cerebral edema and microhemorrhages and is only diagnosable by MRI, making detection and intervention particularly difficult outside of the clinical trial environment [47]. Moreover, in the decades since the amyloid hypothesis was first proposed [20], there have been several studies showing tau may be more closely correlated with cognitive decline [48,49]. This has ultimately led many to question the future of anti-amyloid therapies for the treatment of Alzheimer’s disease. However, these anti-amyloid clinical trials suggest there is still a benefit to targeting amyloid, particularly early in the AD cascade before plaques form, and potentially in combination therapies designed to target multiple AD risk factors.

## 2. Sleep and AD: A Bidirectional Relationship

Sleep changes with normal aging on both a macro- and microstructural level. Older adults have decreased total sleep time and spend less time in deeper stages of non-rapid eye movement (NREM) sleep, or slow-wave sleep (SWS) [50,51,52]. They also have greater sleep fragmentation, leading to increased daytime sleepiness and more frequent napping [53,54]. These age-related changes in sleep are associated with an increased risk of cognitive decline [55,56,57]. In cognitively normal older adults, impaired sleep and, specifically, decreased slow-wave amplitude are associated with increased amyloid and tau burden, particularly in brain regions susceptible to early accumulation of AD pathology [58,59,60,61,62]. Recent studies have also demonstrated the importance of sleep spindles in AD pathophysiology. Impaired coupling between slow waves and spindles, which occurs with normal aging [63], predicted greater tau burden [60], whereas decreased fast spindle density was associated with impaired episodic memory [64]. In both rodent and clinical studies, acute sleep deprivation, specifically deprivation of slow-wave sleep, caused elevated levels of Aβ and tau [65,66,67]. Chronic sleep deprivation increased plaque density and tau propagation and accelerated cognitive decline [65,68,69,70,71]. Several studies have demonstrated that the relationship between sleep and AD is bidirectional, such that disrupted sleep drives AD pathology and the presence of AD pathology reciprocally impairs sleep quality. Adults with MCI or AD spent less time in slow-wave sleep and had decreased sleep efficiency, increased fragmentation, and more frequent daytime naps [13,72,73]. Daily rest-activity patterns are also disrupted in individuals with AD [14,74], and amyloid pathology mediates the relationship between these fragmented activity rhythms and cognitive decline [75]. Sleep disturbances also become more severe as AD progresses, further demonstrating the feedforward loop between AD-related pathology and sleep [73,76,77].

Sleep disruption drives AD pathology through both impaired production and clearance of AD-related proteins. Sleep plays an important role in regulating the diurnal rhythm of Aβ. Both preclinical and clinical studies show Aβ is higher during waking periods and lower during periods of sleep in ISF, CSF, and plasma [68,78,79,80]. This is likely due in part to changes in neuronal firing patterns associated with sleep/wake. SWS is associated with a reduction in global neuronal firing rates and cortical excitability [81,82,83,84,85]. Conversely, prolonged wakefulness is associated with increased brain metabolic activity, neuronal firing, and cortical excitability [85,86,87,88]. Increased neuronal firing is associated with increases in Aβ and tau release [28,65,68], thereby explaining one mechanism through which sleep deprivation contributes to increased circulating levels of Aβ and tau in humans [65,66,89,90]. Slow-wave sleep is also critical for the glymphatic clearance of AD-related proteins from the brain [91,92]. Impaired glymphatic function increases Aβ and tau levels [36,40] while enhancing glymphatic clearance can decrease AD protein levels [93,94]. Both normal aging [95,96] and sleep restriction [97,98,99,100] impair BBB structural integrity, leading to impaired glymphatic clearance and decreased protein clearance. Together, these studies suggest disrupted sleep impairs the balance between AD-protein production and clearance, thereby increasing AD risk.

The presence of AD pathology can also disrupt sleep, creating a feedforward loop in which pathology-related sleep impairments further drive AD pathophysiology. Several well-studied AD mouse models develop sleep disruptions as AD-related pathology accumulates, including increased fragmentation, decreased NREM sleep time, and shifts in power spectra towards higher frequencies [78,101,102,103]. These impairments reflect clinical reports, suggesting these changes in sleep are mediated by the presence of pathology. There are likely several potential mechanisms through which pathology causes sleep disruption. One mechanism is cortical hyperexcitability and subsequent neuronal network dysfunction, which has been demonstrated in both rodent and clinical AD studies [104,105,106,107,108]. Importantly, this hyperexcitability is found in early disease progression, before overt neurodegeneration and hypoactivity [107,108], coinciding with observed sleep disruption [109]. Further, a recent study found age-related hyperexcitability in arousal circuits drove sleep instability and increased fragmentation [110]. While it is unclear if AD exacerbates hyperexcitability specifically in arousal circuits, general hyperexcitability and increased epileptiform activity are well documented in individuals with AD [111]. Interestingly, epileptic activity is associated with both sleep disturbances and exacerbated AD pathology, further supporting the notion that hyperexcitability may play a role in the relationship between sleep and AD progression [112,113]. Several studies have also suggested degeneration of certain neural populations within both wake- and sleep-promoting regions may play a mechanistic role in AD-related sleep fragmentation later in disease progression. AD-related degeneration occurs in several sleep-promoting areas, including galanergic neurons in the ventrolateral preoptic nucleus (VLPO) and melanin-concentrating hormone (MCH) neurons in the lateral hypothalamic area (LHA) [114,115,116]. These neural populations are critical for generating sleep through inhibitory projections to wake-promoting brain regions, and, therefore, their degeneration may contribute to the sleep loss observed in AD patients. There is also degeneration in wake-promoting brain regions, such as the locus coeruleus [117,118], histaminergic neurons in the tuberomammillary nucleus, and orexinergic neurons in the LHA [119,120], potentially contributing to sleep-wake instability and sleep fragmentation.

One final mechanism by which AD pathology may contribute to impaired sleep is through inflammation. Chronic neuroinflammation is a hallmark of AD pathophysiology [121,122,123,124], and studies have shown the sustained presence of activated microglia and reactive astrocytes exacerbates pathology accumulation [125,126,127,128] and impairs protein clearance [129]. Chronic and acute inflammation disrupt sleep [130,131,132], and studies suggest chronic low-grade inflammation may explain some age-associated deficits in sleep [133]. A recent study by Mander et al. found microglial activation mediates the relationship between tau pathology and NREM fast sleep spindle density [64], demonstrating neuroinflammation associated with AD pathology can directly impair sleep function. However, this relationship between inflammation and sleep is bidirectional. Circadian dysfunction, chronic short sleep, and insomnia are all associated with greater inflammation [134,135,136,137], which may be due to shifts in cytokine secretion patterns and increased daytime cytokine levels [138,139], which, in turn, can lead to excessive daytime sleepiness and fatigue [140,141]. Together, these data suggest neuroinflammation may also mediate the bidirectional nature of the relationship between AD and sleep.

## 3. Metabolism and AD

Metabolic syndrome is an independent predictor of mortality [142] and affects nearly 35% of adults in the US, including 50% of adults over 60 years of age [143]. Type 2 diabetes (T2D), hyperglycemia, hyperinsulinemia, and insulin resistance, all factors associated with metabolic syndrome, increase the risk for cognitive and functional decline, frailty, and AD [144,145,146,147,148,149,150,151,152,153]. This elevated AD risk is partially mediated through hypertension and other cardiovascular risk factors common in individuals with metabolic syndrome [154,155,156,157]. However, studies suggest diabetes is an independent risk factor for AD as well [158]. Further, hyperglycemia alone increases ISF Aβ [144] and tau hyperphosphorylation [159,160], suggesting changes in peripheral glucose metabolism may be sufficient to drive AD pathology. In clinical populations, high-glycemic diets led to increased amyloid burden [161,162], while elevated HbA1c levels were associated with AD-related neurodegeneration [163]. Further, imaging studies show that alterations in brain glucose metabolism preceded cognitive symptoms [8,164,165] and predicted cognitive decline [9], suggesting brain glucose metabolism plays a role in early AD pathophysiology. Given this association, several studies have looked at the effect of metabolic syndrome treatments on AD pathology. Preclinical studies found decreased amyloid plaque burden and improved cognitive outcomes after treatment with metformin [166,167], GLP-1 agonists [168,169], and sulfonylureas [170,171]. However, results in clinical studies were varied [172,173,174,175,176,177], indicating the need for further studies to understand the mechanisms behind this relationship.

While there are clear associations between metabolic disruption and AD, the mechanisms by which altered metabolic function drives AD pathology are not clear. One mechanism explored as a potential link between metabolic dysfunction and AD risk is inflammation [178]. Chronic low-grade peripheral inflammation is a hallmark of metabolic syndrome [179,180] and predicts cognitive decline and a more rapid transition from MCI to AD [11,181,182]. It is well documented that chronic peripheral inflammation can lead to neuroinflammation, largely through cytokine-induced disruption of the BBB, which allows a further influx of cytokines into the brain [183]. Studies have demonstrated a similar pattern in metabolic syndrome, where increased permeability of the BBB is associated with neuroinflammation and neurodegeneration [184,185,186]. As previously discussed, neuroinflammation plays a significant role in the pathological progression of AD, exacerbating protein production and impairing clearance. Therefore, peripheral inflammation associated with metabolic syndrome may be a mechanistic link connecting these two diseases [181,187,188].

Disrupted brain insulin signaling and insulin resistance also play a role in mediating cognitive deficits observed with metabolic syndrome [189,190,191]. Moreover, hyperinsulinemia alone has been shown to increase AD risk [192], and the impacts of insulin on AD pathology seem to occur before clinical symptom onset [193]. Hyperinsulinemia increases Aβ production via γ-secretase activity [194], induces Aβ extracellular secretion [195], and increases tau hyperphosphorylation [196]. Hyperinsulinemia also impairs Aβ clearance, as both are degraded by insulin-degrading enzyme (IDE), and, during periods of high brain insulin, IDE will preferentially bind insulin [197,198,199]. There is also evidence that Aβ accumulation can impair insulin signaling and cause brain insulin resistance [200,201]. However, other studies have demonstrated that the brain remains responsive to insulin in the presence of amyloid pathology [202]. In clinical studies, hyperinsulinemic clamps increased Aβ production, but also improved memory [203]. This result led to several clinical trials exploring intranasal insulin as a treatment for cognitive decline [204], which have reported modest cognitive improvement [205,206]. The effect of intranasal insulin might contradict expected outcomes given the associations between hyperinsulinemia and Aβ and tau described earlier. Given these conflicting results, further studies are necessary to understand how insulin contributes to the AD cascade and its future as a treatment option.

## 4. Sleep and Metabolism

The sleep and circadian systems are also major regulators of metabolic activity in the brain and periphery. Blood glucose has a diurnal rhythm, in which glucose levels peak in the evening and glucose tolerance and insulin sensitivity peak in the morning, thus lowering glucose levels [207]. These shifts in blood glucose levels are in part due to decreased glucose utilization during slow-wave sleep [208]. Sleep also regulates the secretion of important hunger- and satiety-related proteins, such as ghrelin and leptin [209,210]. The regulation of these proteins contributes to the diurnal rhythms of glucose homeostasis, as glucose intake follows food consumption patterns. Studies have shown that short sleep and sleep fragmentation are associated with a greater risk of metabolic impairments and an increased prevalence of diabetes [209,211,212,213,214,215]. Specifically, suppression of slow-wave sleep has been shown to impair insulin sensitivity, causing impaired glucose tolerance and a greater risk of metabolic syndrome [213,216]. Therefore, SWS deficits caused by AD pathology may contribute to impaired metabolic function.

Obstructive sleep apnea (OSA) is a common sleep disorder characterized by periods of hypoxemia and/or hypercapnia due to upper airway obstruction, thereby leading to sleep fragmentation. OSA is extremely common in obese individuals and those with metabolic syndrome [217,218,219], and, because of the increasing obesity epidemic, OSA prevalence continues to rise. OSA is also a risk factor for AD, likely because of both the sleep fragmentation and metabolic components [220]. A study by Ju et al. found continuous positive airway pressure (CPAP) treatment for OSA increased SWS, and that increase was associated with decreased Aβ levels [221]. Importantly, CPAP treatment also improved metabolic function [222,223,224]. Together, this suggests sleep improvements may have a beneficial effect on both metabolic status and AD risk.

Sleep disturbances are commonly reported by individuals with T2D and metabolic syndrome [219]. While it is clear that disrupted sleep can impair metabolic functioning, several studies suggest the opposite relationship, whereby metabolic syndrome directly affects sleep, may also exist. A genetic mouse model of diabetes (db/db mice) has increased sleep fragmentation and decreased delta power, a marker of SWS [225]. High-sugar diets are also associated with decreased sleep quality and reduced slow-wave sleep [226,227]. Further, acute instances of hyperglycemia are sufficient to decrease delta power, a marker of SWS, and total NREM sleep time [228]. Several mechanisms might explain this relationship. Hyperglycemia can increase neuronal activity [144,228], which may in turn drive sleep instability and wakefulness [88,110]. Neuroinflammation associated with metabolic syndrome may also disrupt sleep, similar to what is hypothesized to happen with age-related inflammation [133]. While further studies are necessary to identify mechanisms involved, these initial studies suggest a bidirectional relationship between metabolism and sleep, which may contribute to AD pathophysiology. Therefore, when designing AD clinical trials, we should consider multimodal treatment approaches in which multiple AD risk factors are simultaneously targeted. Further, we should prioritize lifestyle interventions in early disease stages and identify those with sleep and metabolic impairments as “at risk” populations, as early targeting of these risk factors will provide the best opportunity for slowing disease progression.

## 5. Conclusions

This review highlights several mechanisms through which disrupted sleep and metabolism, common lifestyle risk factors for AD, contribute to AD pathogenesis and disease progression. We also demonstrate the bidirectional nature of these relationships, where the presence of AD pathology negatively affects sleep and metabolic health, therefore creating feedforward cycles of worsening AD pathology. Finally, we explore the underlying bidirectional connection between these risk factors and suggest potential mechanisms, such as neuroinflammation and neuronal hyperexcitability, by which impaired sleep and metabolic function may synergistically contribute to AD progression. Together, this review highlights the need to identify mechanisms connecting lifestyle risk factors both to AD and to each other, as many of these risk factors are present before clinical AD symptoms appear, and targeting lifestyle risk factors like sleep and metabolic disruption early in disease progression may lower overall AD risk.

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
