# Peer review of "Upsetting the Balance: How Modifiable Risk Factors Contribute to the Progression of Alzheimer’s Disease"

_biomolecules, 2024, doi:10.3390/biom14030274_

Round 1

Reviewer 1 Report

Comments and Suggestions for Authors

First of all I would like to thank the authors for the topic they have selected. Always is interesting to better understand pathogenic mechanisms related to AD pathology, and even more if we talk about risk factors that we potentially can modify.

However, I have some concerns that must be address previous to consider the publication of the work:

1) ABSTRACT: Remove references from the abstract. Include those in the introduction section.

2) Keywords: add the keywords.

3) Introduction:

- add info about epidemiological context of AD.

- reorganize the section talking about biological and clinical definition of AD.

- talk about therapeutic options: not only some of them. Mention the available treatments both approved by EMA and FDA (rivastigmine, memantine...) and later explain the newest ones (but not only Lecanemab or Donanemab, but also Aducanumab). I would suggest to first explain the amylodoigenic hypothesis before starting to explain new anti-amyloid treatments, and consider also to add information how DSAD (Down Syndrome associated Alzheimer's disease) could also support the pathogenic relevance of amyloid.

4) Sleep: I would suggest to create a table summarizing typical sleep modifications in relation to healthy aging and those related with AD pathology. Ideally it could be of interest to compare them with those related with other reasons of cognitive decline (Lewy Body disease, vascular disease...). In turn, I would consider to create a figure that represents the biological mechanisms that link sleep with AD, and ideally not refering only biological repercussions of sleep but also cognitive-behavioral direct influence.

Consider to add more info about the tridirectional relationship between AD-epilepsy-sleep, when talking about the influence of hyperexcitability (is a very hot topic, and I think that should be explained better).

5) Metabolism: again, I would suggest to create table/figure summarizing the main mechanism that link metabolism dysfunction with the main hallmarks of AD and other pahtogenic mechanisms.

Explain differences in metabolism disfunction between those with pure AD or AD-vascular copathology, or only vascular cognitive decline...

Introduce how we can try to modulate this metabolism dysfunction using FINGER like strategies (multidomain lifestyle interventions).

6) Creation of new sections: I would strongly suggest to add the next two sections:

- Differences in relation to sleep and metabolism dysfunction related to AD compared to other reasons of cognitive decine (neurodegenerative and non-degenerative).

- suggestions for clinical assessment of sleep and metabolism dysfunction in the context of cognitive decline; and strategies to prevent/promote healthy lifestyles including those that pretend to reduce sleep disturbances and metabolism dysfunction.

Author Response

We wouuld like to thank thew reivewer for their time and thoughtful comments during this review process. We made the following changes in response to their recommendations: 

  1. Removed all citations from abstract and added them to the introduction where appropriate.
  2. Line 22: Added keywords 
  3. Introduction: We added additional context regarding the epidemiology of AD. We also added additional context on the amyloid hypothesis and treatment options and restructured the section on AD pathophysiology. 
  4. Sleep - We included a graphical abstract that shows age related sleep changes and included both biological and cognitive outcomes. Additionally, we added a few sentences addressing the connection between epileptic activity, sleep, and AD risk. While an in-depth discussion is outside the scope of this opinion paper, we agree that this is an important topic and therefore felt it important to include briefly.
  5. Metabolism - The graphical abstract figure addresses the recommendation for another figure. We also included some lifestyle treatment strategies that we feel would be appropriate to try in future clinical trials to best target both sleep  and metabolic impairments together. We also added some clarification to our section on metabolism regarding vascular pathology. While there are clear connections between metabolic syndrome and vascular dementia, we feel this is appropriately covered by the current literature. We c tohose focus this opinion paper on the interaction between metabolic impairments and AD pathology, indpendent of cardiovascular comorbidities, as these interactions are far less undestood. 
  6. Additional sections: As mentioned, we added some treatment strategies we feel should be tested in future clinical trials addressing these two risk factors in combination. We also added additional clarity to the conclusions to more clearly state we believe these lifestyle interventions should be prioritized as mutifactorial lifestyle treatment approaches. While we did not add an entire new section address sleep and metabolic dysfunction in normal aging, as there are numerous review papers already covering these topics, we did add additional detail to each individual section addressing this. Impaired sleep and metabolic dysfunction are both common in normal aging and can both have negative function outcomes, including impaired cognition, in the absence of any neurodegeneration. We more clearly stated this in the body of the text to reflect the distinction between normal and pathological aging, and added additional context to the graphical abstract as well.  

Reviewer 2 Report

Comments and Suggestions for Authors

In the manuscript titled “Upsetting the balance: how modifiable risk factors contribute to the progression of Alzheimer’s Disease” Caitlin Carroll and Ruth Benca, have described the mechanisms underlying the relationship between disrupted sleep, metabolic dysfunction, and Alzheimer's disease are both timely and insightful. The clarity of writing, the comprehensive coverage of the literature, and the elucidation of bidirectional relationships between lifestyle risk factors and AD pathogenesis are commendable. I believe that this work will make a significant contribution to the field and will be of great interest to the research community and society.

Following are a few minor comments for your consideration:

1.        A graphical abstract illustrating the bidirectional relationship between Sleep and AD and factors influencing these mechanisms, such as metabolism, could be a valuable addition.

2.        Line 18-19: The statement “Sleep and metabolism also appear to have a bidirectional relationship with each other” is redundant with lines 16-17.

3.        Line 27: The abbreviation "ATN" is not described.

4.        Line 32: The abbreviation "MCI" is explained in a later section; consider mentioning it here as well.

5.        Line 57-59: It would be ideal to include a reference for this statement.

6.        Line 103-104: The sentence "The mechanisms through which sleep disruptions drive AD pathology affect both the production and clearance of AD-related proteins" seems incomplete. Please revise for clarity.

7.        Line 180-182: Including a reference for this statement would be ideal.

Thank you for your attention to these minor points. I look forward to the revised version of your manuscript.

Author Response

We would like to thank this reviewer for their comments and time spent on the review. We made the following changes in accordance with their recommendations.  

  1. Added graphical abstract 
  2. Line 18-19: We added additional clarity to these statements to reflect multiple bidirectional relationships exist: (1) sleep and AD, (2) metabolism and AD, and (3) sleep and metabolism.  
  3. Line 46-47: We elaborated further on what each letter in "ATN" refers to. 
  4. Line 50-51: We definted "MCI" at this introduction of the term
  5. Line 62-64: Reference was added
  6. Line 134-135: Clarified sentence
  7. Line 218-219: Reference was added

Round 2

Reviewer 1 Report

Comments and Suggestions for Authors

I thank the authors for including all the sugestions in the last version of the manuscript. I do belive that the paper has significantly improved. 

I do not have further comments.